# The Temporal Dynamics of Sensitivity, Aflatoxin Production, and Oxidative Stress of *Aspergillus flavus* in Response to Cinnamaldehyde Vapor

**DOI:** 10.3390/foods12234311

**Published:** 2023-11-29

**Authors:** Ajuan Niu, Leilei Tan, Song Tan, Guangyu Wang, Weifen Qiu

**Affiliations:** 1Key Laboratory of Grains and Oils Quality Control and Processing, Collaborative Innovation Center for Modern Grain Circulation and Safety, College of Food Science and Engineering, Nanjing University of Finance and Economics, Nanjing 210023, Chinatss1054530271@163.com (S.T.); 2Joint Laboratory for International Cooperation in Grain Circulation and Security, Nanjing 210023, China

**Keywords:** fungi, essential oils, stress response, recovery, AFB_1_

## Abstract

Cinnamaldehyde (CA), a natural plant extract, possesses notable antimicrobial properties and the ability to inhibit mycotoxin synthesis. This study investigated the effects of different concentrations of gaseous CA on *A. flavus* and found that higher concentrations exhibited fungicidal effects, while lower concentrations exerted fungistatic effects. Although all *A. flavus* strains exhibited similar responses to CA vapor, the degree of response varied among them. Notably, *A. flavus* strains HN-1, JX-3, JX-4, and HN-8 displayed higher sensitivity. Exposure to CA vapor led to slight damage to *A. flavus*, induced oxidative stress, and inhibited aflatoxin B_1_ (AFB_1_) production. Upon removal of the CA vapor, the damaged *A. flavus* resumed growth, the oxidative stress weakened, and AFB_1_ production sharply increased in aflatoxin-producing strains. In the whole process, no aflatoxin was detected in aflatoxin-non-producing *A. flavus*. Moreover, the qRT-PCR results suggest that the recovery of *A. flavus* and the subsequent surge of AFB_1_ content following CA removal were regulated by a drug efflux pump and velvet complex proteins. In summary, these findings emphasize the significance of optimizing the targeted concentrations of antifungal EOs and provide valuable insight for their accurate application.

## 1. Introduction

As an infamous fungal species, *Aspergillus flavus* is prone to infecting cereal grains, nuts, and dried fruits, leading to substantial agricultural commodities losses [1,2]. Compounding this issue is the production of hazardous aflatoxins by *A. flavus*, which poses significant risks to both human and animal health [3,4]. Among these aflatoxins, aflatoxin B_1_ (AFB_1_) is the most mutagenic, teratogenic, and carcinogenic mycotoxin, classified as a group I carcinogen by the International Agency for Research on Cancer [5,6]. Therefore, it is necessary to carry out interventions to prevent *A. flavus* growth and aflatoxin contamination in food and feed.

Over the past decades, chemicals have been used to inhibit spoilage microorganisms’ growth and the toxic metabolites’ synthesis. However, this measure has faced substantial criticism due to its significant contribution to drug resistance, adverse effects on human health, and environmental pollution [7,8,9]. Thus, it is now urgent to find and develop environment-friendly alternatives. Essential oils (EOs), a kind of natural plant extract, possess broad-spectrum antimicrobial, insecticidal, and antioxidant properties [10,11,12]. In addition, EOs have been classified as Generally Recognized as Safe (GRAS) by the FDA [13]. Due to the hydrophobicity and low solubility of EOs, the required dose of liquid EOs in practical application is often a higher concentration, which will influence the sensory quality of the final product [14,15]. In contrast, gaseous EO treatment exhibits enhanced antimicrobial effects compared to liquid EOs and offers a promising method to diminish the usage of EOs while preserving their efficacy.

EOs have high volatility, resulting in a gradual decrease in their concentration during application. It is reported that EO treatment intensity, in terms of the time or concentration, affects antimicrobial activities [16,17]. Cinnamaldehyde (CA), a natural aldehyde extracted from plants, possesses prominent antifungal properties and can effectively inhibit both *A. flavus* growth and aflatoxin production [18,19]. Previous research from our team showed that the damage of highly aflatoxin-producing *A. flavus* caused by insufficient CA vapor treatment recovered with the disappearance of CA [20]. It is worth noting that different *A. flavus* strains, especially those aflatoxin-producing and aflatoxin-non-producing strains, exhibit diverse stress-response patterns [17]. Consequently, the regulation mechanisms of growth recovery in different *A. flavus* strains after EO removal also be different. It is necessary to investigate the regulatory responses of different *A. flavus* strains throughout the entire process of gaseous EO treatment, from initial exposure to eventual dissipation.

The objectives of this study were to investigate the sensitivity patterns of different *A. flavus* isolates to CA vapor treatment and analyze the temporal dynamics of aflatoxin production and oxidative stress response. By exploring the expression of relevant genes, we aimed to gain a preliminary understanding of the molecule regulatory mechanisms underlying the response of *A. flavus* to gaseous CA, which can be utilized to optimize the treatment intensities of CA and explore the action target of CA for the mitigation of *A. flavus* and aflatoxin contamination and its associated risks.

## 2. Materials and Methods

### 2.1. Materials

CA and AFB_1_ standards were obtained from Shanghai Yuanye Bio-Technology Co., Ltd. (Shanghai, China). Potato dextrose agar (PDA) was purchased from Qingdao Hope Bio-Technology Co., Ltd. (Qingdao, China). All chemicals were analytical reagents.

### 2.2. A. flavus Isolation and Identification

All *A. flavus* strains were isolated from paddy samples collected from three provinces (Jiangxi, Henan, and Fujian) in southern China. Briefly, aliquots of 10 g of paddy samples and 90 mL of a sterile saline solution were transferred aseptically to a stomacher bag. The mixture was homogenized and shaken for 5 min to prepare spore suspensions. After the prepared spore suspensions underwent 1:10 serial dilutions, the spore suspensions with different concentrations were spread onto PDA plates and cultivated at 28 °C for 7 days. The colonies with yellow and green spores on the PDA plates were selected and purified to obtain single colonies.

The isolates were identified by amplifying and sequencing the calmodulin genes, the primers of which were CL1 and CL2A, as listed in Table 1 [21]. The PCR conditions were: initial denaturation at 94 °C for 5 min, reaction cycles of 94 °C for 30 s, 54 °C for 30 s, 72 °C for 90 s, and final elongation for 7 min at 72 °C. The amplified product was sequenced by Sangon Biotech (Shanghai) Co., Ltd. (Shanghai, China). The sequencing results were then compared against the NCBI database using the nucleotide BLAST program (https://blast.ncbi.nlm.nih.gov/Blast.cgi?PROGRAM=blastn&PAGE_TYPE=BlastSearch&LINK_LOC=blasthome (accessed on 24 October 2023)) to identify *A. flavus* [22].

Inoculum conidia were collected from a 7-day-old fungal strain grown on the PDA plates by adding a sterile 0.85% NaCl solution, and the surface of the mycelium was rubbed using a sterile L-shaped spreader. The final concentration of the spore suspensions was adjusted to approximately 1.0 × 10^6^ spores/mL using a hemacytometer.

### 2.3. Determination of the AFB_1_ Production Ability

The AFB_1_ production ability of the *A. flavus* isolates was measured according to the method of [23], with slight modifications. All *A. flavus* isolates were cultured on PDA plates individually at 28 °C for 7 days. Three 6 mm diameter plugs were excavated from each sample using an Oxford Cup and put into brown centrifuge tubes. Then, 2 mL of high-performance liquid chromatography (HPLC)-grade extraction solvent (methanol/water, 7:3, *v*/*v*) was added. After ultrasonic extraction for 20 min, centrifugation was conducted at 6000 rpm for 10 min, and the supernatant was pipetted into fresh tubes and evaporated by N_2_ flow. The derivatization procedure was carried out by adding 200 μL hexane and 100 μL of trifluoroacetic acid to the residue. The mixture was vortexed for 30 s and left at 40 °C for 15 min. Next, the entire contents continued to be evaporated by N_2_ flow. A mobile phase (methanol/water/acetonitrile, 3:6:1, *v*/*v*) was used to redissolve the residue, and the mixture was vortexed for 30 s. After being filtered by nylon filters (13 mm × 0.22 μm), the filtrate was collected into amber-silanized vials to carry out HPLC analysis. The AFB_1_ standard was dissolved by a mobile phase to prepare a standard solution. A standard curve can be prepared by measuring the AFB_1_ content in standard solutions. The HPLC system used for the AFB_1_ analyses was an Agilent 1260 series system (Agilent Technologies Co., Ltd., Palo Alto, CA, USA) equipped with a fluorescence detector (FLD). The excitation wavelength and emission wavelength were, respectively, 365 nm and 464 nm. The stationary phase was a C18 column (Phenomenex Luna 5 μm, 150 × 4.6 mm). The mobile phase mentioned above was used, the flow was 1 mL/min, and the run time was 15 min.

### 2.4. CA Vapor Sensitivity Testing

Certain isolates were specifically chosen to encompass a diverse range of *A. flavus* with varying abilities to produce AFB_1_. Four representative isolates of aflatoxin-producing *A. flavus* (JX-3, JX-4, FJ-5, and HN-8) and four representative isolates of aflatoxin-non-producing *A. flavus* (JX-1, JX-2, FJ-4, and HN-1) were selected for the subsequent test. Aliquots of 10 μL *A. flavus* spore suspension were inoculated on PDA plates by the point inoculation method and incubated at 28 °C for 3 days. Then, filter papers containing CA were pasted on the inner cover of Petri dishes. The concentrations of the CA vapor were, respectively, 0, 0.1, 0.2, 0.4, 0.6, 0.8, and 1.6 μL/mL, which were speculated by dividing the volume of the remaining air in the Petri dishes by the amount of CA. Parafilm (Bemis, Neenah, WI, USA) was used to seal the plates. After 12 h of incubation, the morphological changes and colony diameters of the recovered *A. flavus* were recorded to assess the CA vapor sensitivity of the *A. flavus* strains. The treatment groups with less recovered hyphae and slower growth rates were considered to be more sensitive.

The recovery of *A. flavus* was determined according to the method described by [24], with minor modifications. After 12 h of treatment, new sterile covers were used to take the place of the covers containing the antifungal atmosphere, and the *A. flavus* continued to cultivate for 3 days. Morphological changes in the *A. flavus* at different stages were observed and photographed. The groups with and without CA vapor treatment were treated as the control (C) and treatment (T) groups, respectively, while the groups treated with cover displacement were treated as the recovery (R) group.

The colony diameters were determined according to the method of Wang et al. [25], with minor modifications. Treated and untreated spores were acquired by washing the plates with sterile saline solutions. Then, 10 μL spore suspensions were added to the PDA plate centers and then cultured at 28 °C. The colony diameter was determined every day during the incubation period (7 days) using the cross method.

### 2.5. AFB_1_ Production Analysis after CA Vapor Treatment

Spore suspensions (10 μL) were inoculated in the center of the PDA plates. After 3 days of incubation, CA was dripped onto sterile filter papers taped onto the plate cover’s inner center. Based on the results of the sensitivity testing, a CA concentration of 0.4 μL/mL was selected for subsequent experiments to explore the damage and recovery of different *A. flavus* strains in response to CA. These plates were wrapped with parafilm and cultured for 12 h. Then, the plate covers were replaced with sterile covers, and incubation was continued for 3 days. The AFB_1_ contents in the C, T, and R groups were measured as described above.

### 2.6. Evaluation of Oxidative Stress Response

#### 2.6.1. Measurement of Reactive Oxygen Species (ROS)

The level of ROS was detected following the directions of the Reactive Oxygen Species Assay Kit (Beyotime Biotechnology, Shanghai, China). Firstly, the mycelia in the C, T, and R groups were stained with a 10 μM DCFH-DA probe. After incubation at 37 °C for 20 min, the mycelia were washed with PBS three times, fixed on slides, and then their fluorescence intensity was observed with a fluorescent microscope. The fluorescence intensity represents the contents of ROS.

#### 2.6.2. Measurement of Catalase (CAT) and Superoxide Dismutase (SOD) Activities

*A. flavus* mycelia were homogenized in cold PBS (0.01 mol/L and pH = 7.4) and centrifuged at 15,000× *g* for 10 min at 4 °C to obtain the supernatant [16]. According to the instructions of the Catalase Assay Kit and Total Superoxide Dismutase Assay Kit with WST-8 (Beyotime Biotechnology, Shanghai, China), CAT and SOD in the supernatant were determined by a spectrophotometer at wavelengths of 520 nm and 450 nm. The activity of CAT was determined by calculating the amount of H_2_O_2_ catalyzed by CAT in the sample converted into H_2_O and O_2_ in units of time and units of volume. The activity of SOD had a negative correlation with the amount of formazan dye converted by WST-8, which was obtained by the colorimetric analysis of a WST-8 product.

#### 2.6.3. Measurement of Malondialdehyde (MDA) Content

A Lipid Peroxidation MDA Assay Kit (Beyotime Biotechnology, Shanghai, China) was used based on the manufacturer’s protocol to determine the content of MDA. The reaction between MDA and thiobarbituric acid (TBA) produces red products. The absorbance was measured by a spectrophotometer at a wavelength of 532 nm, and the MDA content was calculated through a standard curve.

### 2.7. RNA Extraction

The total RNA was extracted by the Trizol method. Mycelia growing for three days and continuing to grow for three days after CA removal were collected and grounded with liquid nitrogen. Then, 1 mL of Trizol reagent was added to the mycelia and incubated for 15 min. Then, 200 μL chloroform was added and vortexed 15 s. After 5 min of incubation, centrifugation was conducted at 4 °C, 12,000 rpm for 15 min. Following centrifugation, the mixture was divided into three phases, and the upper transparent aqueous phase contained the RNA. An aliquot of 400 μL of aqueous phase was transferred to a fresh RNase-free tube, mixed with 400 μL isopropyl alcohol, and placed for 10 min. The RNA precipitate, which is often invisible before centrifugation, formed a pellet on the side and bottom of the tube after centrifugation. Removing the supernatant, the RNA pellet was mixed with 1 mL of 75% ethanol and recollected by centrifugation at 4 °C, 10,000 rpm for 5 min. This step was repeated once. At the end of the procedure, the RNA pellet was first dried and then dissolved in RNase-free water.

The concentration and purity of the RNA were appraised using a NanoDrop 2000 spectrophotometer (Thermo Scientific, Waltham, MA, USA). The integrity of the RNA was appraised by agarose gel and a gel imaging system (Tianneng Instrument Co., Ltd., Shanghai, China).

### 2.8. qRT-PCR Analysis

Reverse transcription and qRT-PCR were both carried out using a kit (Vazyme Biotech Co., Ltd., Nanjing, China) according to the manufacturer’s directions. The reaction for reverse transcription was conducted at 37 °C for 15 min and then at 85 °C for 5 s. Totals of 10 μL 2 × ChamQ SYBR qPCR Master Mix, 2 μL cDNA, 0.8 μL primer, and 7.2 μL RNase-free ddH_2_O existed in the qRT-PCR reaction volumes. The specific primers used in the qRT-PCR are listed in Table 1. The qRT–PCR program included an initial denaturation at 95 °C for 30 s and cyclic reactions of 95 °C for 10 s, 60 °C for 30 s, 95 °C for 15 s, 60 °C for 60 s, and 95 °C for 15 s. The 2^−ΔΔCt^ method was used to determine the relative quantification of a target gene by comparing it with a reference gene [26].

**Table 1 foods-12-04311-t001:** Primers used for qRT-PCR.

Gene	Forward Primer (5′–3′)	Reverse Primer (5′–3′)	Reference
*Actin*	ACGGTGTCGTCACAAACTGG	CGGTTGGACTTAGGGTTGATAG	[27]
*CL1/CL2A*	GARTWCAAGGAGGCCTTCTC	TTTTTGCATCATGAGTTGGAC	[21]
*laeA*	GAAAGGTTGCTCGCTGGTA	GAACGCCTCCGACTTGACT	This study
*velB*	GTAGACTTGTGGAACGCAGAG	AGAGGACATAGCCGTGGAT	This study
*vosA*	GTGGGAAAGAGAAAGAACGC	GCAGCACATAAAATAATAGGGACT	[18]
*fnx1*	AGGCAAGTCTCCGAGTGAA	CCGAAGATTAGCCAAAACC	This study
*mdrA*	TTGCTTGTGTGCCTTTTCCCTT	TCCCCAAATCCTGTCCTCCAT	This study
*FLU1*	ATTCTTGGCTTCGCTTTTGGA	GCGGCGGTATTCTTGCTTGTT	This study

### 2.9. Statistical Analysis

All data were recorded as the mean ± SD of three independent replicates. Statistical analyses and graph drawing were conducted using GraphPad Prism 9.4 software. Student’s *t*-test and one-way ANOVA were used to evaluate the differences among the data in the different groups. Differences were considered significant at *p* < 0.05.

## 3. Results

### 3.1. Fungal Strain Identification

The *A. flavus* isolates were initially screened by their morphology, further confirmed by comparing the calmodulin sequences of these isolates using the BLAST algorithm, and identified as *A. flavus* strains. The AFB_1_ production ability of the *A. flavus* isolates was determined by HPLC. According to these results, these isolates can be categorized into aflatoxin-producing *A. flavus* and aflatoxin-non-producing *A. flavus*. Among them, *A. flavus* JX-1, JX-2, FJ-4, and HN-1 were unable to produce this aflatoxin, while *A. flavus* JX-3, FJ-5, HN-8, and JX-4 were capable of producing AFB_1_, and their production rates were, respectively, 902.02, 256.03, 133.263, and 100.88 ng/mL (Figure 1). *A. flavus* JX-3 showed the highest AFB_1_ production ability.

### 3.2. CA Vapor Sensitivity Patterns of A. flavus

All *A. flavus* strains were sensitive to CA vapor, with their growth being inhibited in a concentration-dependent manner. With the increase in the concentration of CA vapor, the damage degree of *A. flavus* increased, resulting in a slower growth rate of the treated spores (Figure 2a,b, Appendix A). In most cases, the colony diameter of the spores treated with 0.4 μL/mL CA was significantly different from that of the untreated spores and spores treated with 0.8 μL/mL CA vapor during the same incubation time. After being treated with a gaseous CA concentration of 1.6 μL/mL, all spores of *A. flavus* lost their ability to grow. It was observed that different isolates of *A. flavus* presented varying sensitivities to gaseous CA. Specifically, after being treated with a CA concentration of 0.6 μL/mL, the spores of *A. flavus* HN-1, JX-3, and JX-4 did not exhibit any growth on the first day, indicating a higher level of sensitivity compared to the other five isolates of *A. flavus*.

The morphological changes in the *A. flavus* strains are shown in Figure 2c,d, Appendix A. When the CA vapor was removed, the previously inhibited *A. flavus* resumed growth. Notably, all isolates of *A. flavus* recovered after CA removal when the concentration of the CA vapor was below 0.4 μL/mL. At a concentration of 0.4 μL/mL, all *A. flavus* recovered slightly. Meanwhile, an observation of the colony sizes revealed that *A. flavus* JX-3, JX-4, HN-1, and HN-8 showed relatively poor recovery ability. After treatment with 0.6 and 0.8 μL/mL gaseous CA, partial recovery of *A. flavus* was observed. CA vapor at a concentration of 1.6 μL/mL completely inhibited the growth of *A. flavus*. Among the tested concentrations, the experimental results were found to be the most consistent and stable when the concentration of CA vapor was set at 0.4 μL/mL. Thus, considering the sensitivity of eight *A. flavus* isolates to CA comprehensively, 0.4 was selected as the CA concentration for subsequent experiments.

### 3.3. AFB_1_ Production Analysis after CA Vapor Treatment

Gaseous CA treatment had an inhibitory effect on AFB_1_ synthesis in aflatoxin-producing *A. flavus* JX-3 and JX-4, as evidenced by a decrease in the AFB_1_ levels compared to the control group. However, after the removal of the CA vapor, a sharp increase in AFB_1_ synthesis was observed in these strains (Figure 3), which were as high as 1439 ng/mL and 1347 ng/mL, respectively. AFB_1_ of aflatoxin-producing *A. flavus* FJ-5 and HN-8 was not detected in either the group in which *A. flavus* grew for three days or the group in which the *A. flavus* was continuously treated with CA vapor for 12 h. During the entire period of the CA vapor treatment and removal, aflatoxin-non-producing *A. flavus* JX-1, JX-2, FJ-4, and HN-1 did not produce AFB_1_.

### 3.4. Evaluation of Oxidative Stress Response in A. flavus

The responses of aflatoxin-producing *A. flavus* and aflatoxin-non-producing *A. flavus* to oxidative stress were similar. Fluorescence microscope observation showed that the green fluorescence of *A. flavus* in the T group was obvious and continuous. (Figure 4 and Appendix A). In contrast, the fluorescence intensity of *A. flavus* was weaker in the R group, whereas the mycelia were not stained in the C group.

The activities of CAT and SOD (Figure 5 and Appendix A) were found to be increased in the T group as compared to the C group. The activities of these antioxidant enzymes in *A. flavus* were reduced after CA removal. Likewise, gaseous CA caused a substantial increase in the MDA levels in *A. flavus* cells (Figure 5 and Appendix A). The content of MDA returned to normal levels in the R group.

### 3.5. Effects of CA Vapor on Velvet Complex Proteins and Drug Efflux Pump Gene Expression

The gene expression profiles of velvet complex proteins were down-regulated in both the aflatoxin-producing and aflatoxin-non-producing *A. flavus*, while the expressions of three velvet complex protein genes at different *A. flavus* strains had varied levels (Figure 6). Gene expression increase was determined in terms of the drug efflux pumps. However, there was wide variation in the expression levels of the drug efflux pumps in different *A. flavus* strains, and in some instances, the expression of certain genes was not detected.

## 4. Discussion

EOs possess excellent antimicrobial properties against food-related microorganisms, which have been widely reported [28,29,30]. Depending on the specific doses utilized, EOs exhibit fungicidal or fungistatic effects on fungi [31,32,33]. Due to their high volatility, the concentration of EOs will gradually decrease or even disappear during the course of actual usage. Therefore, impaired fungi can recover growth after the elimination of a low-intensity EO treatment. Růžička et al. [34] reported that the growth of yeasts was completely inhibited when the concentrations of monocaprin were 150–200 mg/L, while the growth of the most filamentous fungi was inhibited at concentrations of 100–400 mg/L. Consistent with these findings, this study observed that a high concentration of CA vapor completely prevented the growth of *A. flavus*, while *A. flavus* treated with low-concentration CA vapor recovered growth after CA removal. These results were found in both aflatoxin-producing and aflatoxin-non-producing *A. flavus* strains. The observed effect persisted regardless of the removal of the antifungal environment or the transfer of spores to a new environment. Therefore, the inhibitory effect of gaseous CA on *A. flavus* growth could be attributed to the interference of the metabolism process of *A. flavus*. However, it is worth noting that the sensitivity pattern of *A. flavus* to CA vapor varied from different *Aspergillus* strains, among which *A. flavus* JX-3, JX-4, HN-1, and HN-8 showed high sensitivity to CA vapor.

Based on previous research, EOs have demonstrated the ability to inhibit mycotoxin production [35,36,37]. However, these studies have primarily paid more attention to the persistent treatment process and have overlooked the possibility of the incomplete eradication of fungi. It is important to consider that residual fungal presence may pose a continued risk of mycotoxin contamination. In this work, gaseous CA treatment was found to effectively inhibit the production of AFB_1_. Nevertheless, the synthesis of AFB_1_ in all aflatoxin-producing *A. flavus* strains increased sharply once the CA vapor was removed. Therefore, it is of great importance to completely inhibit the growth of *A. flavus* in practical applications by using high-intensity EO treatment. In addition, aflatoxin was not detected in aflatoxin-producing *A. flavus* FJ-5 and HN-8 after three days of growth. By contrast, *A. flavus* JX-3 and JX-4 were capable of accumulating aflatoxin during the early growth stage, thereby amplifying the potential harm associated with mycotoxin contamination.

The velvet complex, formed by a combination of *veA*, *velB*, *vosA*, and *laeA*, serves as a global regulator in the secondary metabolism of fungi [38,39]. Kale et al. [40] found that the deletion of *Laea* in *A. flavus* generated the deficiency of aflatoxin synthesis, with restoration observed in complemented strains. Bayram et al. [41] demonstrated the absence of detectable aflatoxin production in *Aspergillus nidulans* strains with deletions of either *velB* or *veA*. Wang et al. [42] reported a decrease in aflatoxin content with increasing expression levels of the *laeA* and *velB* genes in *A. flavus* treated with CA. Consistent with previous findings, a good correlation was observed between the increased AFB_1_ content and the downregulation of the *laeA*, *velB*, and *vosA* genes. It is worth noting that aflatoxin-non-producing *A. flavus* also showed the same gene expression pattern. Previous research by our team found no correlation between the sharp increase in AFB_1_ in recovered *A. flavus* JX-3 and the two key regulators, *aflR* and *aflS*, involved in aflatoxin biosynthesis [18]. Therefore, it is comprehensively predicted that the velvet complex influences the structural genes of AFB_1_ synthesis through global regulation, leading to the elevation of aflatoxin production.

A burst of ROS is often regarded as a sign of apoptosis and can easily disturb cellular oxygen metabolism and induce oxidative stress [43,44]. Studies have reported that oxidative stress in cells may enhance the activity of antioxidant enzymes, which in turn alleviates the damage of ROS to cells [45,46]. CAT and SOD are important antioxidant enzymes involved in cellular defense mechanisms. SOD converts superoxide radicals to H_2_O_2_, which is then decomposed by CAT [47]. In the present study, gaseous CA treatment led to the accumulation of ROS, accompanied by a corresponding increase in antioxidant enzyme activities. The observed variations in the ROS levels and antioxidant enzyme response were found to be associated with the differences among the *A. flavus* strains and treatment conditions. The heightened CAT and SOD activities observed in the T group suggest activation of the defense system against oxidative damage. After the CA vapor was removed, the ROS in the *A. flavus* significantly decreased, while antioxidant enzyme activities returned to normal levels, suggesting the cessation of the defense mechanism. In the same group, the levels of ROS contents and antioxidant enzymes among different *A. flavus* strains were also different. Cells often undergo membrane lipid peroxidation when exposed to a large amount of ROS [48,49]. MDA, as one of the important products of lipid peroxidation [45,50], is also produced subsequently. The substantial increase in the MDA levels following the gaseous CA treatment further confirms the induction of oxidative stress in *A. flavus* cells. The subsequent normalization of MDA content in the R group indicates the restoration of cellular homeostasis.

Previous studies have shown that fungi can diminish drug levels by activating drug efflux transporters and enhancing the exogenous detoxification ability to alleviate unfavorable growth situations [51,52]. The drug efflux pumps in fungi are divided into two categories: significant facilitator superfamily transporters (MFS) and ATP-binding cassette (ABC) transporters [53]. It was reported that a large proportion of genes belonging to MFS were up-regulated in *Penicillium digitatum* treated by the compound nanoemulsion [54]. In this study, the expressions of genes *FLU1*, *mdrA*, and *fnx1*, which are related to drug efflux pumps, were examined by qRT-PCR. On the whole, the colony diameter of *A. flavus* in the recovery group was positively correlated with the up-regulation of the drug efflux protein gene. For example, the colony was larger in the recovered aflatoxin-producing *A. flavus* JX-4 and FJ-5 and the aflatoxin-non-producing *A. flavus* FJ-4 and JX-1. Therefore, it can be inferred that the recovery of *A. flavus* after the removal of CA could be ascribed to the persistent up-regulation of multidrug resistance gene expression.

Through the above experimental findings, it was observed that all strains of *A. flavus* showed comparable responses to the existence and disappearance of CA vapor. However, the levels of tolerance to oxidative stress between the aflatoxin-producing and aflatoxin-non-producing *A. flavus* strains were different, and generally aflatoxin-producing *A. flavus* strains tolerate higher oxidation levels [55,56]. For example, Narasaiah et al. [57] showed that the activities of xanthine oxidase and free radical scavenging enzymes (SOD and GSH-Px) in aflatoxin-producing *A. flavus* increased compared to the aflatoxin-non-producing strain in the presence of H_2_O_2_. However, Fountain et al. [58] found that most toxigenic isolates were able to tolerate higher concentrations of H_2_O_2_ in toxin-conducive media, while there were also certain toxigenic isolates that exhibited comparable tolerance levels to those of the atoxigenic isolates in their study. Probst et al. [59] found that the addition of H_2_O_2_ to aflatoxin-non-conducive YEP medium induced the production of ROS, while a decrease in aflatoxin production did not significantly impact the level of ROS. Similarly, in this study, some aflatoxin-producing *A. flavus* possessed strong levels of oxidative stress tolerance and damage repair capacity, while others demonstrated weaker responses. Therefore, it is likely that the extent of tolerance to oxidative stress caused by CA among *A. flavus* isolates could also be modulated by additional crucial factors unrelated to aflatoxin production, which needs to be further studied.

## 5. Conclusions

In this study, the sensitivity and response patterns of different *A. flavus* isolates to CA vapor were investigated. It was found that high-intensity CA vapor treatment exerted fungicidal effects, while mild treatment exerted fungistatic effects and allowed for growth recovery after CA vapor removal. CA vapor revealed an inhibitory effect on AFB_1_ synthesis. Exposure to CA vapor also led to the accumulation of endogenous ROS in *A. flavus*, along with increased activity of antioxidant enzymes and the content of MDA. However, upon the removal of CA vapor, there was a sharp increase in the AFB_1_ content in the recovered *A. flavus*, and the levels of ROS and MDA, as well as the activities of CAT and SOD, returned to their original levels. Notably, different *A. flavus* strains exhibited similar changes in the oxidative stress induced by CA vapor, albeit with varying degrees of response. Furthermore, the utilization of qRT-PCR highlighted the regulatory role of drug efflux proteins and velvet complex proteins in the recovery of *A. flavus* and the subsequent sharp increase in AFB_1_ content following CA removal. These results emphasize the importance of applying a sufficiently intense treatment of EOs to completely inhibit the recovery of *A. flavus* and the subsequent increase in mycotoxin production, which provide valuable theoretical guidance for the practical application of EOs in controlling *A. flavus* contamination and mycotoxin formation.

## Figures and Tables

**Figure 1 foods-12-04311-f001:**
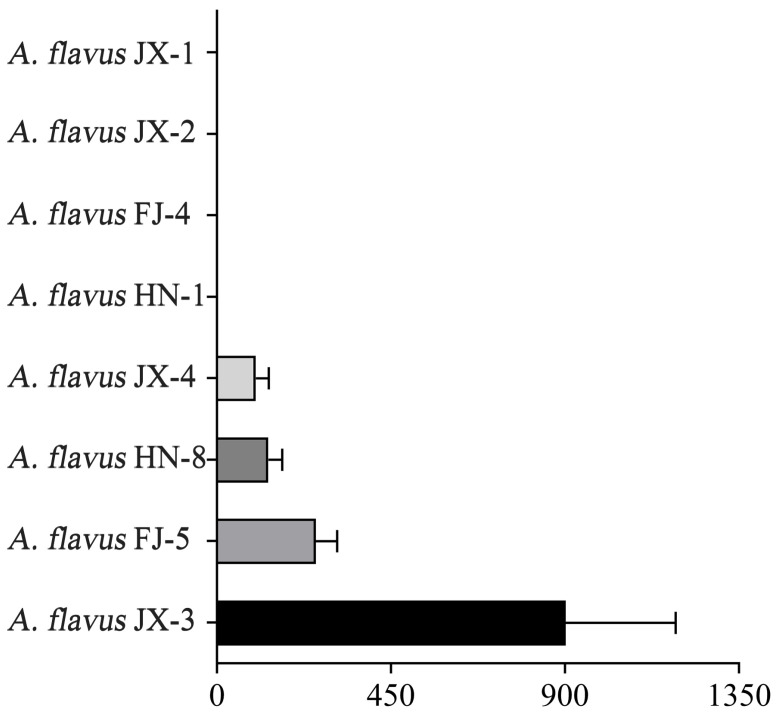
Columns with different lengths represent the production of AFB_1_.

**Figure 2 foods-12-04311-f002:**
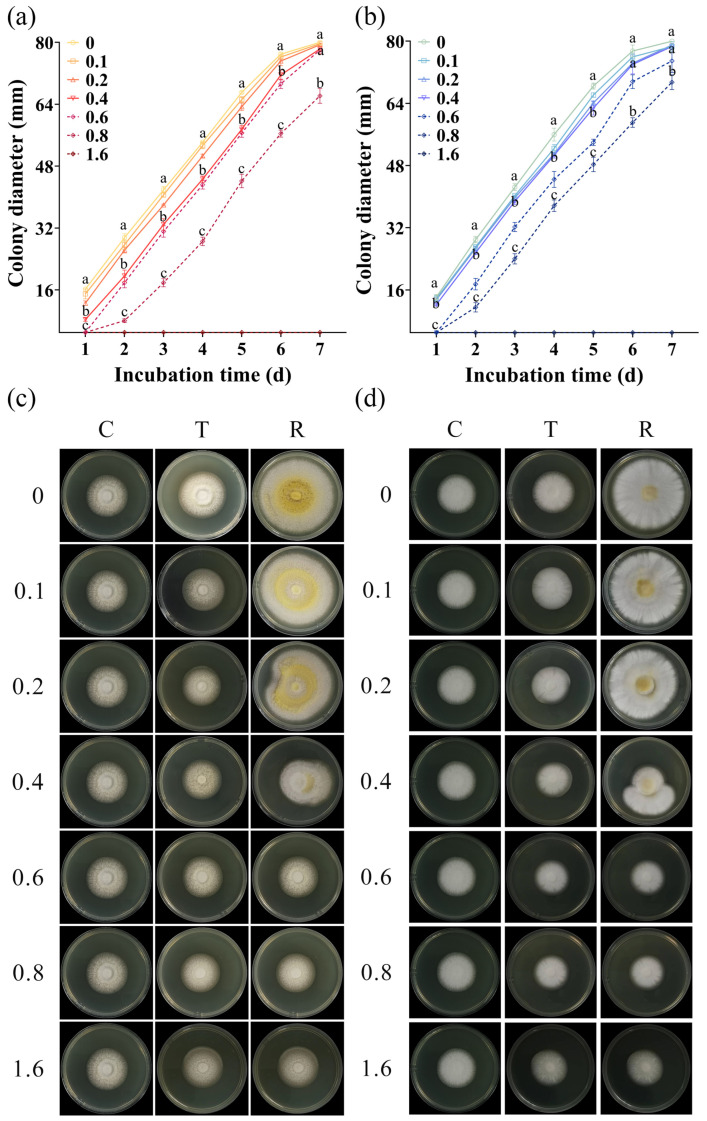
CA vapor sensitivity patterns representative of *A. flavus*. Growth curves (**a**,**b**) and morphological changes (**c**,**d**) of aflatoxin-producing *A. flavus* JX-3 and aflatoxin-non-producing *A. flavus* HN-1. C—control group, T—group treated with CA vapor for 12 h, R—group with CA vapor that disappeared after 12 h treatment. The concentrations of the CA vapor were 0, 0.1, 0.2, 0.4, 0.6, 0.8, and 1.6. The colony diameters of spores treated by 0, 0.4, and 0.8 μL/mL CA vapor were selected as typical representatives. Values in the same incubation time with different lowercase letters are significantly different (*p* < 0.05).

**Figure 3 foods-12-04311-f003:**
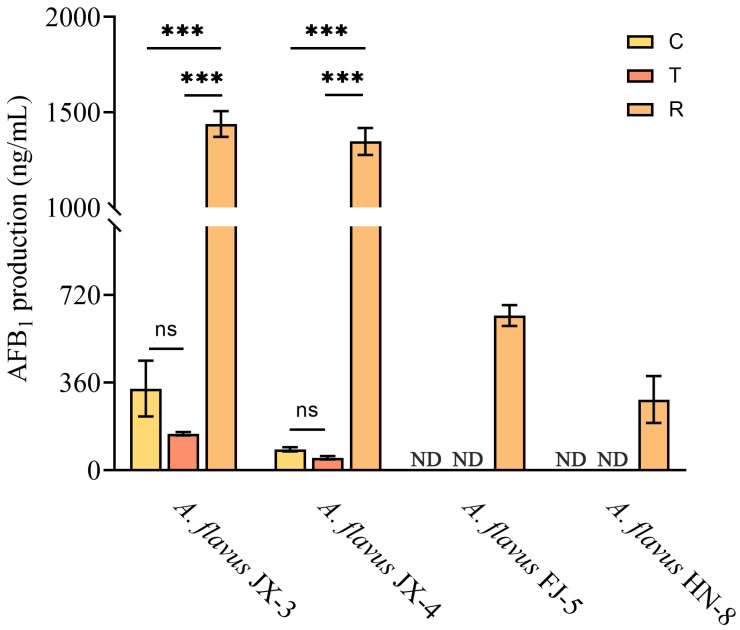
AFB_1_ production analysis after CA vapor treatment in aflatoxin-producing *A. flavus*. ND indicates that AFB_1_ was not detected in this group; ns—no significant differences. *** *p* < 0.001. C—control group, T—group treated with CA vapor for 12 h, R—group with CA vapor that disappeared after 12 h treatment.

**Figure 4 foods-12-04311-f004:**
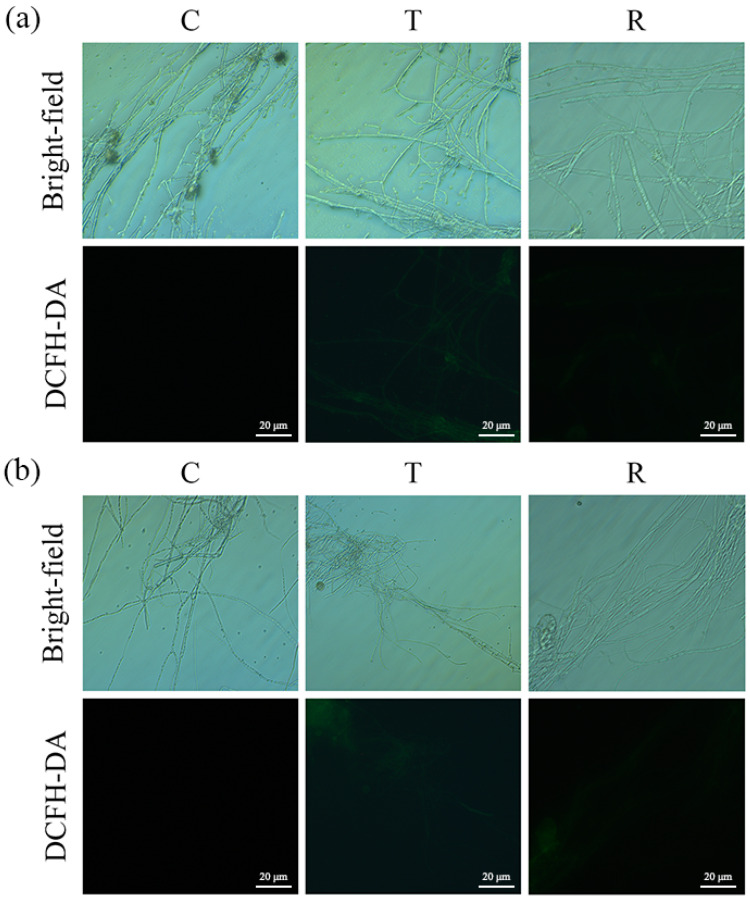
Changes in ROS in *A. flavus* JX-3 (**a**) and *A. flavus* HN-1 (**b**). C—control group, T—group treated with CA vapor for 12 h, R—group with CA vapor that disappeared after 12 h treatment.

**Figure 5 foods-12-04311-f005:**
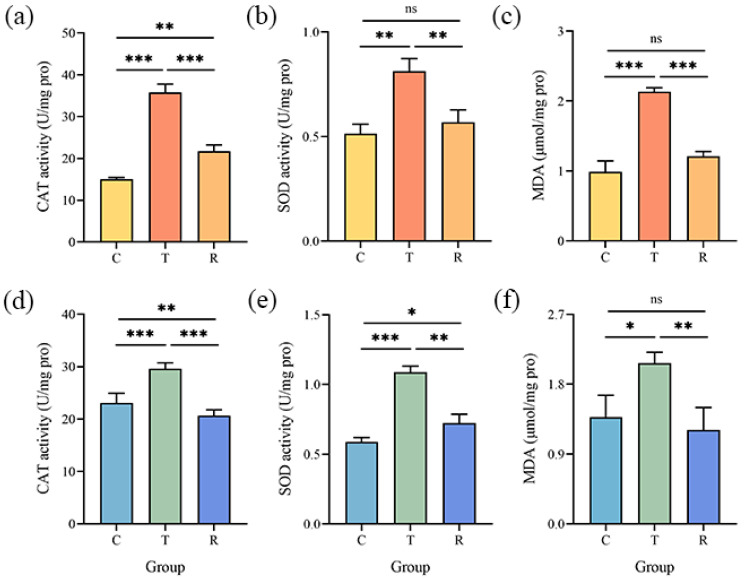
Changes in CAT (**a**,**d**), SOD (**b**,**e**) activities, and MDA content (**c**,**f**) in *A. flavus* JX-3 (**upper panel**) and *A. flavus* HN-1 (**lower panel**); ns, no significant differences. * *p* < 0.05, ** *p* < 0.01, *** *p* < 0.001. C—control group, T—group treated with CA vapor for 12 h, R—group with CA vapor that disappeared after 12 h treatment.

**Figure 6 foods-12-04311-f006:**
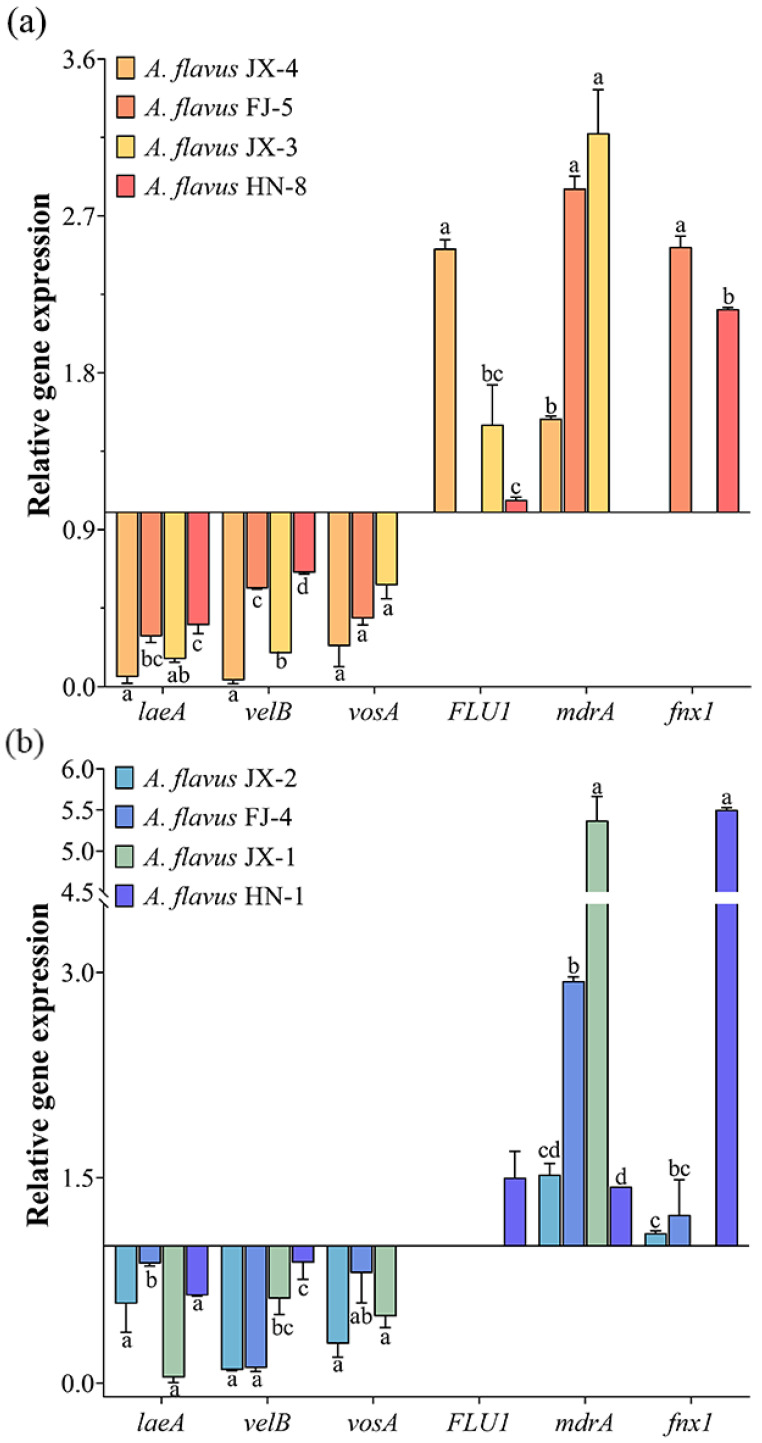
Effects of CA vapor on velvet complex proteins and drug efflux pump gene expression in aflatoxin-producing *A. flavus* (**a**) and aflatoxin-non-producing *A. flavus* (**b**). Different lowercase letters indicate significant differences (*p* < 0.05).

## Data Availability

The data used to support the findings of this study can be made available by the corresponding author upon request.

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
