# Peer review of "The Temporal Dynamics of Sensitivity, Aflatoxin Production, and Oxidative Stress of *Aspergillus flavus* in Response to Cinnamaldehyde Vapor"

_foods, 2023, doi:10.3390/foods12234311_

Round 1
Reviewer 1 Report
Comments and Suggestions for Authors
Given below is a critical analysis highlighting these issues:
The manuscript lacks a clear structure, with sections often blending into one another. Reorganize the content to have a more logical flow, with clear section headings.
The abstract should be more concise. It should provide a brief overview of the objective of the study, methods, key findings, and implications. The abstract should be structured to give readers a quick summary of the main points of the study.
The introduction lacks a clear statement of the objectives or hypotheses of the study. It should specify what the researchers aimed to achieve with this study. The introduction provides background information but does not discuss prior research related to the specific topic. This omission limits the context and relevance of the study. The manuscript needs to explain the relevance of the findings of the study more explicitly. Why is it important to understand the sensitivity of A. flavus to CA vapor, and how can these findings be applied in practice?
In “Materials and methods”, the section describing the isolation and identification of A. flavus isolates is insufficient. It should include details of the sampling process, isolation procedures, and validation methods.
The description of the AFB1 production analysis should include more details on the procedure, reagents used, and equipment.
The section on CA vapor sensitivity testing needs to clarify how concentrations were determined and how sensitivity was assessed.
The information on the evaluation of oxidative stress response should include a description of the methods and reagents used to measure ROS, CAT, SOD, and MDA.
The RNA extraction and qRT-PCR analysis should be described more comprehensively, including primer sequences and cycling conditions.
The results section should provide a detailed presentation of the data and their implications.
The discussion section should interpret the results in the context of existing literature and provide perceptions into the significance of the findings.
The discussion should address the limitations of the study, the broader implications of the results, and potential future research directions.
The section on the correlation between oxidative stress and drug efflux pumps gene expression is vague and should provide more detailed explanations.
The conclusion section summarizes the key findings and their implications.
Overall, the manuscript needs substantial revision to address these issues and provide a more thorough and coherent presentation of the research findings. Additionally, it should include clear implications and applications of the research results for the scientific community or practical use.
Comments on the Quality of English LanguageThe manuscript contains numerous grammatical and typographical errors, and the text lacks clarity in several places. It would benefit from thorough proofreading and editing for language and structure.
Author Response
- The manuscript lacks a clear structure, with sections often blending into one another. Recorganize the content to have a more logical flow, with clear section headings.
Response: Thank you for your valuable comments concerning our manuscript. We have thoroughly checked, summarized and revised the content of the manuscript to improve its flow and readability.
- The abstract should be more concise. It should provide a brief overview of the objective of the study, methods, key findings, and implications. The abstract should be structured to give readers a quick summary of the main points of the study.
Response: Thank you for your valuable comments. We have revised the abstract (line 11-19, 20-24).
- The introduction lacks a clear statement of the objectives or hypotheses of the study. It should specify what the researchers aimed to achieve with this study. The introduction provides background information but does not discuss prior research related to the specific topic. This omission limits the context and relevance of the study. The manuscript needs to explain the relevance of the findings of the study more explicitly. Why is it important to understand the sensitivity of flavus to CA vapor, and how can these findings be applied in practice?
Response: Thank you for your valuable comments. We have revised the introduction (line 43-47, 53-57, 63-67).
- In “Materials and methods”, the section describing the isolation and identification of flavus isolates is insufficient. It should include details of the sampling process, isolation procedures, and validation methods.
Response: We have rewritten this part and clearly describe the isolation and identification of A. flavus isolates (line 74-81, 83-86 ).
- The description of the AFB1 production analysis should include more details on the procedure, reagents used, and equipment.
Response: Thank you for your valuable comments. We have revised it (line 103-106, 109-110).
- The section on CA vapor sensitivity testing needs to clarify how concentrations were determined and how sensitivity was assessed.
Response: Thank you for your valuable comments. We have revised this section (line 121-127).
- The information on the evaluation of oxidative stress response should include a description of the methods and reagents used to measure ROS, CAT, SOD, and MDA.
Response: Thank you for your suggestion, we have revised this part (line 152-155, 158-165, 168-171).
- The RNA extraction and qRT-PCR analysis should be described more comprehensively, including primer sequences and cycling conditions.
Response: Thank you for your comments, we have listed the primer sequences in Table 1. The cycling conditions were listed in line 195-197.
- The results section should provide a detailed presentation of the data and their implications.
Response: Thanks for your suggestion. We have revised it (line 211-215, 222-226, 254-256).
- The discussion section should interpret the results in the context of existing literature and provide perceptions into the significance of the findings.
Response: Thanks for your suggestion. We have revised the discussion (line 304-310, 314-318, 327-332, 381-391).
- The discussion should address the limitations of the study, the broader implications of the results, and potential future research directions.
Response: Thanks for your suggestion. We have revised the discussion (line 304-310, 314-318, 327-332, 381-391).
- The section on the correlation between oxidative stress and drug efflux pumps gene expression is vague and should provide more detailed explanations.
Response: Thanks for your suggestion. We have complimented some reference in this section (line 381-386).
- The conclusion section summarizes the key findings and their implications.
Response: Thanks for your suggestion. We have summarized the conclusion (line 393-397, 399-402, 405-409).
- Overall, the manuscript needs substantial revision to address these issues and provide a more thorough and coherent presentation of the research findings. Additionally, it should include clear implications and applications of the research results for the scientific community or practical use.
Response: Thanks for your suggestion. We thoroughly checked and revised the whole contents of the manuscript.
Reviewer 2 Report
Comments and Suggestions for Authors
The topic is promising, focusing on the impact of CA on the growth and AFB1 production in various strains of A. flavus. Nevertheless, some modifications are needed.
1.Rewrite the statement "Although all A. flavus …….. were more 16 sensitive. " (Lines 15-16) as “While all A. flavus strains exhibited similar responses to CA vapor, the degree of response varied among them. Notably, A. flavus strains HN-1, JX-3, JX-4, and HN-8 displayed higher sensitivity."
2.Rewrite the key words. Avoid the use of keywords such as “Aspergillus flavus; Cinnamaldehyde; Sensitivity; Aflatoxins” that already appeared in the title.
3.The basic drawback of the EO application is its volatile nature. Please address this issue and in support of the findings of the present study.
4. In section 2.2 mention the PCR conditions and about the sequencer.
5.Identities are missing in the last clad of the phylogenetic tree (figure 1). The graph representing in the figure 1 concerning the AFB1 quantification lack the SD. IS it a single value or the mean value of three independent experiments.
6.Apply the statistical difference in the colony diameter graphs with different treatment of CA and incubation time (figure 2 A and B).
7.Provide the scale bar in the figure 4. Apply the statistical difference between the groups in the graphs represented in figure 6.
Comments on the Quality of English Language
Moderate english editing
Author Response
- Rewrite the statement "Although all A. flavus …….. were more 16 sensitive." (Lines 15-16) as “While all A. flavus strains exhibited similar responses to CA vapor, the degree of response varied among them. Notably, A. flavus strains HN-1, JX-3, JX-4, and HN-8 displayed higher sensitivity."
Response: Thanks for your suggestion. We have rewritten this part (line 14-16).
- Rewrite the key words. Avoid the use of keywords such as “Aspergillus flavus; Cinnamaldehyde; Sensitivity; Aflatoxins” that already appeared in the title.
Response: Thank you for your valuable comments. We have rewritten key words (line 25).
- The basic drawback of the EO application is its volatile nature. Please address this issue and in support of the findings of the present study.
Response: Thank you for your valuable comments. Considering your suggestion, we have revised discussion and conclusion (line 304-310, 405-409).
- In section 2.2 mention the PCR conditions and about the sequencer.
Response: Thank you for your suggestion, we have complemented the PCR conditions (line 83-86) and the sequencers (line 86).
- Identities are missing in the last clad of the phylogenetic tree (figure 1). The graph representing in the figure 1 concerning the AFB1 quantification lack the SD. IS it a single value or the mean value of three independent experiments.
Response: Thank you for your valuable comments. The main purpose of this study is to identify these strains as different A. flavus strains by caM sequencing and morphological characteristics, and to study their different sensitivity patterns to CA vapor. Therefore, we deleted the content of constructing a phylogenetic tree in order to focus on the research topic. As for the AFB1 quantification the result is the mean value of three independent experiments and we have revised figure 1.
- Apply the statistical difference in the colony diameter graphs with different treatment of CA and incubation time (figure 2 A and B).
Response: Thank you for your suggestion, we have selected colony diameters of spores treated by 0, 0.4 and 1.6 μL/mL CA vapor as typical representatives, and the significance of colonies caused by the concentration of CA vapor was analyzed (figure 2 A and B, figure S1 A and B, figure S2 A and B, figure S3 A and B).
- Provide the scale bar in the figure 4. Apply the statistical difference between the groups in the graphs represented in figure 6.
Response: Thank you for your suggestion, we have complimented scale bar in the figure 4, S4, S5 and S6 and applied the statistical difference between the groups in figure 6.
Round 2
Reviewer 1 Report
Comments and Suggestions for Authors
The only remaining issues are minor grammatical errors, which can easily be rectified during the proofreading of the final galley if the manuscript is accepted.
Comments on the Quality of English LanguageThe only remaining issues are minor grammatical errors, which can easily be rectified during the proofreading of the final galley if the manuscript is accepted.
Reviewer 2 Report
Comments and Suggestions for Authors
The MS has been substantially revised. I will recomend to publish the artilce.
Comments on the Quality of English LanguageNA